# Cyanoacrylate Adhesives for Cutaneous Wound Closure

**DOI:** 10.3390/ani14182678

**Published:** 2024-09-14

**Authors:** Inácio Silva Viana, Paula Alessandra Di Filippo, Francielli Pereira Gobbi, Rachel Bittencourt Ribeiro, Gabriel João Unger Carra, Luiza Maria Feitosa Ribeiro, Lara de Souza Ribeiro, Michelle do Carmo Pereira Rocha, Paulo Aléscio Canola

**Affiliations:** 1Clinical and Animal Surgery Laboratory, Faculty of Agricultural and Veterinary Sciences, São Paulo State University “Júlio de Mesquita Filho”, Jaboticabal 14884-900, São Paulo, Brazil; gabriel.carra@unesp.br (G.J.U.C.); mc.rocha@unesp.br (M.d.C.P.R.); 2Clinical and Animal Surgery Laboratory, Sciences and Agricultural Center Technologies, State University of the North Fluminense (UENF), Campos dos Goytacazes 28013-602, Rio de Janeiro, Brazil; pdf@uenf.br (P.A.D.F.); franci_gobbi@hotmail.com (F.P.G.); f.feitosaribeiro@gmail.com (L.M.F.R.); 3Pathology and Morphology Animal Laboratory, Sciences and Agricultural Center Technologies, State University of the North Fluminense (UENF), Campos dos Goytacazes 28013-602, Rio de Janeiro, Brazil; rachel_bittencourt@hotmail.com (R.B.R.); lara.ribeiro@uenf.br (L.d.S.R.)

**Keywords:** inflammation, collagen, surgery, healing, resistance

## Abstract

**Simple Summary:**

This study investigated the use of cyanoacrylate-based adhesives and their effects on the closure of surgical wounds. Different commercial and surgically indicated cyanoacrylate molecules were compared with each other and with conventional suture. The materials were evaluated for polarization time, scar resistance, inflammatory index, and healing. Promising results were observed in surgical wounds treated with adhesive compared to suture thread.

**Abstract:**

Cyanoacrylate-based adhesives are widely used in wound closure, providing good cosmetic results and little discomfort. However, reports in the literature are found about negative effects that include the release of cytotoxic chemicals during biodegradation. In this study, we sought to evaluate and compare the effectiveness of four cyanoacrylate-based adhesives on the closure of skin incisions in *Rattus norvegicus*. The animals (*n* = 140) were divided into five groups of 28 animals each according to the wound closure technique: G1 and G2 (n-2-ethyl-cyanoacrylate); G3 (n-2-butyl-cyanoacrylate); G4 (n-2-octyl-cyanoacrylate); and G5 (5 nylon stitches). Midline incisions measuring 5.0 cm in length were created and closed using the different materials evaluated, and on D3, D7, D14, and D21, tensiometric and histopathological analyses were performed. Shorter wound closure and adhesion times were observed in G4 animals. At D3 and D7, G5 presented greater tensiometric resistance in the animals of G5, with a decrease in D14 and D21 compared to the other groups. On the other hand, the wounds of G3 and G4 were more resistant in D14 and D21, reaching maximum resistance values. Polymorphonuclear and mononuclear cells are more prevalent and more granulation tissue was observed in G5. The deposition of type III collagen was more evident in G5, whilst there was no difference in the amount of type I collagen in any of the groups treated with cyanoacrylate adhesives. Larger areas stained positive for VEGF-α in G2 and smaller areas in G4, with peaks at D7 and D14. In general, cyanoacrylate adhesives cause less intense inflammatory reactions, resulting in shorter healing times when compared to nylon sutures.

## 1. Introduction

The surgical approximation of tissues is necessary for the recovery of the shape and function of an injured organ or anatomical region. A major challenge faced by surgeons is the need to minimize surgical time and postoperative complications, resulting from inadequate tissue closure [1]. This problem can be ameliorated by using substances that enable faster and more efficient tissue closure. These substances must have a hemostatic effect, adhere firmly to the affected tissues without altering the healing process, and cause no negative side effects nor have carcinogenic action [2]. Adhesive materials are a viable alternative to traditional sutures because they cause less pain on application, are faster and easier to apply, and cause less scarring [3].

Adhesives currently available for this purpose include fibrin combined with thrombin, fibrin-resorcin-formaldehyde, and collagen combined with thrombin. Adhesives based on cyanoacrylate are attractive for surgical use due to their wide availability and low cost, in addition to their strong adherence to tissues, rapid hardening, instant adhesion, and ease of use [3,4]. Cyanoacrylate is a liquid with low density that polymerizes into long chains in the presence of weak bases such as water and blood, forming a strong film on the application surface that can join the wound edges [4]. A number of cyanoacrylate molecules are available, differing in the length of the side chain: methyl (R=CH_3_), ethyl (R=C_2_H_5_), butyl and isobutyl (R=C_4_H_9_), octyl-cyanoacrylate (R=C_8_H_17_), and decyl-cyanoacrylate (R=C_10_H_21_) [3].

In human medicine, adhesives based on cyanoacrylate are commonly used in children because they cause less pain and do not require bandages. However, they are also used in dentistry, orthopedics, plastic surgery, video surgery, mastology, thoracic surgery, ophthalmology, and head and neck surgery, with promising results for emergency wound closure [5]. The use of adhesives has also been described in wild animals, with the advantage of avoiding the need for manipulation for daily bandage change and minimizing the stress on the animal, as well as for use in ophthalmic and otological wound closure [6]. 

Most experimental studies have focused on characterizing the inflammatory index caused by different suture materials or between adhesives and sutures. However, little is known about the differences in healing resulting from the application of cyanoacrylate adhesives containing short and long polymeric carbon chains. Thus, the present study sought to compare the effect of four different cyanoacrylate-based topical adhesives (n-2-ethyl-cyanoacrylate, n-2-butyl-cyanoacrylate, and n-2-octyl-cyanoacrylate) with nylon thread in the closure of an experimental cutaneous surgical wound in rats (Rattus norvegicus) (Figure 1).

## 2. Material and Methods

This study was approved and supervised by the institution’s animal use and care committee, under protocol number 2016-326. 

### 2.1. Experimental Animals

A total of 140 Wistar rats (*Rattus norvegicus*) were used in the study. All rats were healthy uncastrated males and more than 10 weeks of age. During the study, the animals were kept in individual cages under controlled environmental conditions (23 ± 1 °C, with a relative humidity of 55 ± 5% and 12-h light–dark cycle), with water available ad libitum in stainless steel troughs and specific rodent feed (Presence^®^, Cravinhos, Brazil). 

### 2.2. Wound Creation

For the creation of the cutaneous wounds, the animals were anesthetized with a combination of 10% ketamine hydrochloride (Syntec^®^, Tambore, Brazil), at a dose of 100 to 200 mg/kg and 2% xylazine hydrochloride (Syntec^®^, Tambore, Brazil) at a dosage of 5 to 16 mg/kg, with both being administered intraperitoneally (IP). After anesthesia, the haircoat was clipped from the thoracoabdominal region and antiseptic preparation with 2% chlorhexidine gluconate and 70% alcohol was performed. A craniocaudal cutaneous incision approximately 5.0 cm in length was made over the midline using a scalpel and a 24-blade. 

### 2.3. Experimental Design

After making the surgical incision, the animals were randomly allocated into one of five groups, with each group consisting of 28 animals, according to the closure method and adhesive used, as follows: G1: n-2-ethyl-cyanoacrylate (Three bonder^®^, Cincinnati, OH, USA); G2: n-2-ethyl-cyanoacrylate (Super bonder^®^, New York, NY, USA); G3: n-2-butyl-cyanoacrylate (Histoacryl^®^, Caretersa, Spain); G4: n-2-octyl-cyanoacrylate (Liqui Band^®^ Rapid, Plymouth, UK); and G5: five simple sutures with nylon thread (Technofio^®^, Goiania, Brazil). In animals treated with the adhesives, immediately after the incision was made, the edges were approximated and the adhesives were applied topically along the incision line. In G5 animals, the wounds were closed with a 4-0 thread, using single interrupted sutures, spaced 0.5 cm apart. The sutures were removed on the 14th day after the surgical procedure.

For analgesia, the animals were given dipyrone (Dorfin^®^, Mogi Mirim, Brazil), 160 mg/kg, subcutaneously, every 12 h (q12h), for three days. No topical wound therapy was applied in any of the groups. 

### 2.4. Sample Collection and Analysis

The analysis began intraoperatively (D0), with the closure times (juxtaposition of the wound edges) and film formation being measured with a stopwatch.

On days 3, 7, 14, and 21, seven animals from each group were euthanized by administration of pentobarbital (Pisabental^®^, Salvador, Brazil), 100 to 200 mg/kg, IP, followed by the collection of skin samples for histopathological and tensiometric evaluation.

For tensiometric analysis, skin tissue (integument) samples measuring 3 × 7 cm were collected from five animals, immediately immersed in a sodium chloride (NaCl) solution, and sent for analysis in a DL 20000 testing machine. The wounds were fixed in claws attached by aluminum rods aligned parallel to the incision and the claws were connected to the machine, which applied traction perpendicular to the incision line at a speed of 50 mm/min. Tensile strength was measured automatically and constantly during the test. The results were measured in kilograms of force to achieve rupture and converted into grams of force.

The samples were carefully dissected and fixed in 10% formalin buffer, embedded in paraffin, cut with a microtome to obtain 5-µm thick sections (Leica Biosystems RM2245, Heidelberger, Germany), and stained with hematoxylin and eosin (H/E) and Picrosirius red. For immunohistochemistry, after initial histotechnical processing, heat-induced epitope retrieval pretreatment was carried out in a water bath with anti-VEGF-α monoclonal primary antibody (Cat #: ab1316, Abcam, Cambridge, UK) at a dilution of 1:400 and polymer label detection system (Novolink Max Polymer Detection System, Leica Biosystems, Deer Park, IL, USA).

Histological sections were evaluated under an Eclipse 80i microscope (Kurobane Nikon Co., Otawara, Tochigi, Japan) using the NIS Elements BR software (New York- USA) to analyze the presence of inflammatory infiltrate of polymorphonuclear and mononuclear cells, granulation tissue, type III and I collagen fibers, and VEGF-α. To evaluate inflammatory infiltrate and granulation tissue, a score was assigned: 0: absent; 1: mild, 2: moderate; and 3: strong [7,8]. This analysis was performed blindly by three experienced pathologists. The presence of type III, type I collagen, and VEGF-A was determined by the percentage of positive area using ImageJ V1.52 software (National Institutes of Health, Bethesda, MD, USA) used for image analysis.

In two animals from each group, at each time point, the adhesive was removed to evaluate the surgical wound.

### 2.5. Statistical Analysis

All data were expressed as mean ± SD for all variables. Descriptive statistics were calculated for all variables. The data were classified according to the nature of the aspects evaluated and subjected to tests: Student’s T-tests, One-way ANOVA, Tukey’s parametric, and Mann–Whitney’s non-parametric tests. All analyses were performed using the R software (R Core Team, Vienna, Austria) with a significance level of 5% (*p* < 0.05).

## 3. Results 

The time for incision closure was three times longer in G5 animals than in the other four groups (Figure 2). Of the groups with adhesive application, G4 had the shortest closure time. A transparent film was formed on the wounds by all synthetic adhesives (G1–G4). On D14, these films were no longer visible in animals from G1 and G2, while on D21, they only continued to be visible in animals from G4. In general, the adhesive films were brittle (Table 1).

Rupture at the incision site occurred in all samples submitted to the resistance test, highlighting the fragility and low resistance of the wounds during the healing process. On D3 and D7, skin samples from the group with wounds closed with nylon thread (G5) had higher resistance than those treated with adhesives (G1, G2, G3, and G4). Conversely, on D14 and D21, samples treated with adhesives were more resistant than those that had been sutured (G5). Sutures were removed on D14. When comparing the groups in which adhesives were employed, G3 (D14) and G4 (D21) showed greater resistance than the other groups (Figure 3).

The evolution of the scarring process, as well as the intensity of polymorphonuclear, mononuclear, and giant cells and the proliferation of granulation tissue, are shown in Table 2 and Figure 4.

The presence of a polymorphonuclear inflammatory infiltrate (predominantly neutrophils) was observed in all groups, especially in D3 and D7 in all groups, with a subsequent decrease at evaluation times D14 and D21. The greater polymorphonuclear infiltrate was observed in G5 compared to the other groups, with no statistical differences being observed between the groups in which the adhesives were applied, although lower values were observed in G3 and G4.

In all groups, the mononuclear infiltrate, mainly macrophages and lymphocytes, increased from D3 to D14 but then decreased until the end of the study. The mononuclear infiltrate was less evident in G4, compared to G5, at all evaluation times. Differences were also found in G1 and G3 in relation to G5 from D7 to D21.

Giant cells were observed in all groups, with an increase in intensity from D3 onward; however, there were no significant differences between treatments at any of the times evaluated.

Granulation tissue was identified with marked neovascularization, fibroblasts, and loose connective tissue. The greatest amount of granulation tissue was observed from D7 to D21 in G5 animals. There was no significant difference between the other groups, despite the lower values obtained in G3 and G4 in relation to the other groups.

As for fibroplasia, there was a peak in type III collagen between D7 and D14, while type I collagen was more evident on D21. The values obtained through field filling at 400× magnification are shown in Table 3. In Figure 5, the evolution of collagen can be seen, as well as the filling of the surgical wound over time (at 40× magnification). The presence of type III collagen was more evident in G5 at all times compared to the other groups. In relation to type I collagen, differences were observed on D14 and D21 between treatments; G4 followed by G3 presented higher total values than the other groups.

In relation to VEGF-α (Figure 6A), immunohistochemistry showed significant differences in the percentage of immunostained areas in the samples between the groups at all evaluations. A high percentage of expression was seen as early as the third postoperative day, with peaks at different times of evaluation between the groups (Table 3). Significant differences were observed on D3 between G5 and other groups, as well as G1 and G2 in relation to G3 and G1, and G2 and G3 in relation to G4. On D7, lower percentages were quantified in G4 and G3 compared to G1 and compared to G2 and G5, respectively. Such differences remained until D14. Finally, on D21, higher expression of VEGF-α was present in G5 and G1 compared to G3 and G4.

In the macroscopic evaluation of the wound and its evolution (Figure 6B), it was observed that healing in the groups treated with cyanoacrylate adhesives was practically complete on D7, while in the group with conventional sutures, the incision was still quite evident. There was no difference in healing responses between groups from D14 onward. These observations were not subjected to statistical tests, since only two animals were evaluated at each time in each group.

## 4. Discussion

The cyanoacrylate polymer has a low-density liquid state that, when in contact with anionic substances, quickly polarizes, forming a film on the applied surface [9]. Due to this characteristic, the time for cutaneous wound closure in the groups treated with cyanoacrylate adhesives was substantially reduced compared to the time for conventional suturing. This resulted in a shorter intraoperative period, a desirable outcome for unstable patients or those with cardiovascular disorders, in addition to a lower risk of infection and shorter inhalation anesthesia time, minimizing dose- and time-dependent effects on myocardial and sympathoadrenal activity [10]. The shortest closure time was observed in G4 (n-octyl-cyanoacrylate), followed by G3 (n-butyl-cyanoacrylate). However, we should emphasize that this study only considered the adhesive application time in 5 cm incisions and the maintenance of the juxtaposition of the wound edges after application. Similar findings were described in the fixation of abdominal mesh in rabbits, where butyl showed slower polarization than hexyl and octyl, but octyl also showed slower polarization than hexyl; a second finding is attributed to the addition of inhibitors, thickeners, and plasticizers used in manufacturing that can change the polarization time of the molecule [11]. Adhesives with larger carbon chains tend to have greater intramolecular interactions, quickly triggering polarization of the molecule in the presence of anionic substances or structures such as blood or body tissues, while methyl and ethyl molecules are less polarized and more electronegative and consequently require more energy and time for polarization [12,13,14].

The greater resistance of conventional sutures over adhesives is due to the anchoring of the thread to the tissue, while with adhesives, this adhesion is easily lost, causing slippage between the adhesive and the integument [15]. Similar findings were described [16] in a study of sheep tracheas, where sutures were compared with n-butyl-cyanoacrylate adhesive. However, in another study, evaluating the tensile strength by volumetric pressure between thread, fibrin glue, and cyanoacrylate adhesive in end-to-end enteroanastomosis in pigs, the thread and fibrin glue showed inferior results to the cyanoacrylate adhesive [17]. In our study, the suture showed superior results up to D7, with a decrease in tensiometric resistance compared to adhesives after this. Regarding the resistance between adhesives, similar results were described in the evaluation of the resistance of wounds induced on the back of rabbits, with an increase in resistance over time, especially in wounds treated with adhesives with larger carbon chains [5,18,19,20]. These reports corroborate the findings of this study, where adhesives with larger carbon chains (butyl and octyl-CA) had greater resistance compared to ethyl-CA. Adhesives with shorter carbon chains have less flexibility, making them more fragile and brittle [5], as observed in G1 and G2, which were treated with ethyl-cyanoacrylate. Cracks in the adhesive were also observed in G3 and G4, but the material was more flexible and adhered to the skin for longer compared to G1 and G2. The cyanoacrylate monomer is composed of carbon, cyano, and alkyl groups, the latter two being highly electronegative and responsible for the rigidity of the film formed when polarized [9,14] In monomers with smaller carbon chains, there is a higher concentration of these compounds, giving greater rigidity to the monomer [14]. Radicals that incorporate the growth of polymer chains have reduced intramolecular forces due to the number of carbon atoms, which in turn form linear chains of stable anionic bonds with the skin, promoting greater elasticity and durability of the adhesive [5].

In the study of wound closure materials, many criteria are evaluated such as cytotoxicity, degradation, resistance, and foreign body reaction [21]. Thus, it is a common consensus that the number of inflammatory cells present in an injured tissue will depend on the severity of the injury and the closure technique [22]. The scores attributed to edema, granulation tissue, polymorphonuclear, mononuclear, and giant cells and their arrangement in the lesion bed are similar to those around the material used throughout the postoperative evaluation. In this study, the histological findings of the wounds, in which different cyanoacrylate and nylon thread adhesives were used, in general, showed a proliferative inflammatory process and early maturation, respectively, compatible with inflammation induced by surgical incision.

Neutrophils are the first defense cells targeted to an injury; neutrophil chemotaxis becomes more prolonged to allow these cells to perform their function of phagocytosis or destruction of microorganisms in response to reaction to foreign bodies through enzymatic mechanisms, low O_2_ tension, decreased pH, and presence of reactive nitrogen and oxygen species [3,9]. In this study, there was a higher incidence of neutrophils in the inflammatory phase (D3), with a consequent decrease during the subsequent evaluations. The group in which suture thread was used for closure (G5) had the highest scores for neutrophil infiltration, with such findings being associated with tissue damage triggered by the passage of the needle, as well as the irritation caused by the permanent suture material between the edges of the wound. Adhesives form an occlusive film without any trauma during application or foreign body effects. 

Unlike neutrophils, macrophage numbers remained high, with a slight decrease over time in all treatments. Significant differences were also found in G5 in relation to the other treatments and, like neutrophils, the high values of these cells are associated with the degree of injury. Inflammatory patterns with a greater infiltrate of mononuclear cells have already been described in studies that evaluated the healing of surgical wounds with topical application of smaller carbon chain molecules in rats [9,23,24]. However, despite the groups being treated with applications containing a lower number of carbon chains, there were no significant differences between the different molecules. Macrophage infiltration occurs through the signaling of microenvironment molecules in endothelial cells that cause altered expression of surface molecules facing the vascular bed to facilitate the adhesion of circulating macrophages in preference to neutrophils after the initial inflammatory process [25,26,27]. In addition, macrophages play a fundamental role in debridement and their major contribution relates to the secretion of cytokines and growth factors that stimulate angiogenesis, fibroplasia, and extracellular matrix synthesis, which is also associated with the degree of injury [24,27]. 

Furthermore, studies show that cyanoacrylate has antimicrobial properties against Escherichia coli, Staphylococcus aureus, Staphylococcus epidermidis, Corynebacterium pseudodiphtheriticum, Klebsiella pneumoniae, Pseudomonas aeruginosa, and Staphylococcus aureus [24,25]. Such properties may be beneficial in relation to the inflammatory infiltrate in wounds closed with CA adhesives. It should be noted that no post-surgical dressings were applied in any of the groups.

The similarity in granulation tissue deposition and organization observed between the adhesive and suture groups is in agreement with the findings of previous reports comparing the compatibility of ethyl cyanoacrylate and butyl cyanoacrylate adhesives with nylon thread in the repair of incisions on the back of rats [3]. Granulation tissue formation is mediated by transforming growth factors beta and consists of a provisional matrix, including high levels of fibronectin, hyaluronic acid, and type III collagen, which is vital for cell adhesion and migration to the granulation tissue that is crucial for filling the wound area, homeostasis, cell migration and adhesion, and scar resistance [27].

Additionally, the shorter alkyl chain cyanoacrylate adhesive degrades in a shorter time when compared to long-chain monomers, resulting in the deposition of the tissues of a greater quantity of degradation products (cyanoacetate and formaldehyde) which, consequently, triggers inflammatory reactions and more intense histotoxicity [4,28,29]. In this study, there was a trend of a higher incidence of inflammatory cells in wounds treated with ethyl-cyanoacrylate in relation to wounds treated with butyl and octyl-cyanoacrylate, but there was no significant difference. Furthermore, despite the cytotoxicity mentioned in relation to the molecule applied in G1 and G2, inflammatory infiltrates were statistically lower in general compared to G5 at all evaluation moments.

The deposition and reorganization of components in the connective tissue in the wound healing process will affect the resistance and conformity in the repair. Type III collagen shows production spikes during the inflammatory phase; over time, the initial collagen (type III collagen) is reabsorbed and a thicker collagen (type I collagen) is produced and organized along the tension lines during the proliferative phase, with an inverse relationship in type III and I collagen deposition according to the healing phase [27]. A study carried out with rabbits in which the efficiency of surgical closure was evaluated with butyl cyanoacrylate and suture thread [30] observed that the adhesive had a lower inflammatory index, and subsequently, there was less deposition of type III collagen, while type I collagen was similar between the groups, giving greater resistance to healing. In another study evaluating the application of butyl-cyanoacrylate, hexyl-cyanoacrylate, octyl-cyanoacrylate, and polyamide thread in the repair of abdominal hernia in rabbits, it was observed that animals treated with adhesives showed less deposition of type III collagen when compared to thread, and the opposite situation was observed in the deposition of type I collagen [19]. These results corroborate the findings of the present study, in which the differences between the groups were associated with the intensity of the inflammation triggered by each material used. The rats in groups treated with octyl-cyanoacrylate and butyl-cyanoacrylate adhesives showed less deposition of type III collagen when compared to animals where wounds were closed with surgical thread, while type I collagen was similar between groups.

Regarding the expression of VEGF-α (Table 3), a greater variation in positive areas was observed, with significant differences between the groups. Greater expression of positive areas was observed in G5 and lower in G4 at all times evaluated. The inflammatory infiltrates are multifunctional secretory cells of the immune system that regulate the immune response, through the release of chemical mediators such as cytokines and growth factors in the face of an injury, to promote healing [27]. VEGF-α is one growth factor, which has the effect of decomposing the extracellular matrix, stimulating migration and proliferation of endothelial cells proportional to the inflammatory stimulus [31]. This effect is shown in Figure 3, with a greater degree of granulation tissue and consequently vessels present in G5.

Finally, although this study brings relevant results regarding the use of cyanoacylate-based adhesives in the synthesis of surgical wounds and healing, there is still no consensus on the use of these polymers. Questions regarding the cytotoxicity, exothermic effect, and activity of stabilizing substances on the effects and polarization of these monomers are still the subject of constant questioning, and there is a need for studies to elucidate these issues.

## 5. Conclusions

Topical use of cyanoacrylate adhesive has been shown to be a beneficial alternative to sutures for closing skin wounds in low-tension areas. Those containing longer carbon chains (butyl and octyl cyanoacrylate) promote a less intense and more organized inflammatory reaction, with greater collagen deposition and greater tensile strength during healing, thus requiring less time for the healing process. However, even cyanoacrylate polymers with shorter chains showed promising results regarding the elements evaluated when compared to traditional suture thread.

## 6. Contribution to the Field Statement

The increasing prevalence of accidents and surgical procedures involving domestic animals raises the need to evaluate the benefits of new tissue closure materials. Synthetic adhesives are increasingly being used in human and veterinary medicine. This study demonstrated, through resistance tests, the calculation of the inflammatory index and observation of collagen deposition and the viability and efficacy of different cyanoacrylate monomers versus traditional skin sutures. In addition to being easy to apply, it also presents a shorter time for wound closure and consequently, a reduction in trans-operative time.

## Figures and Tables

**Figure 1 animals-14-02678-f001:**
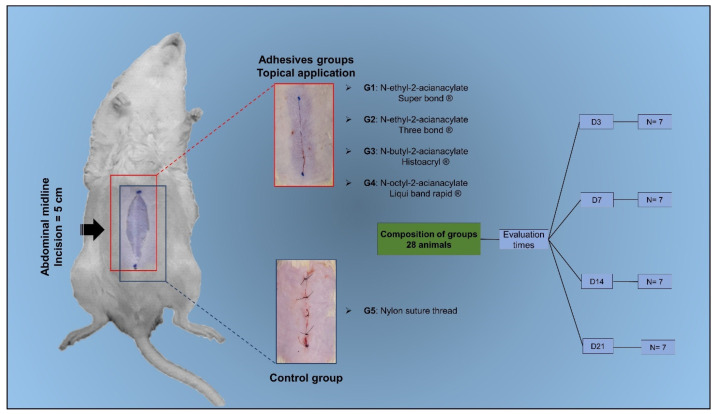
Schematic diagram of surgical wound closure using cyanoacrylate-based adhesives and conventional sutures (nylon thread) evaluated at different times (3, 7, 14, and 21 days postoperatively).

**Figure 2 animals-14-02678-f002:**
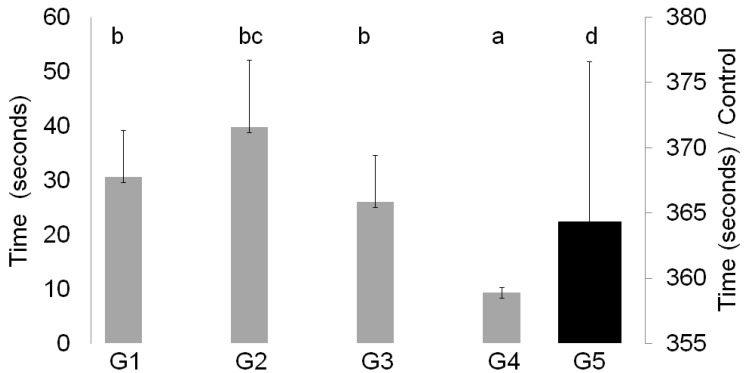
Mean ± standard deviation of wound closure time (seconds) in animals from groups G1, G2, G3, G4, and G5. Means followed by distinct lowercase letters indicate differences between groups by the Tukey test at *p* < 0.05% probability. G1: n-2-ethyl-cyanoacrylate (Three bonder^®^); G2: n-2-ethyl-cyanoacrylate (Super bonder^®^); G3: n-2-butyl-cyanoacrylate (Histoacryl^®^); and G4: n-2-octyl-cyanoacrylate (Liqui Band^®^ Rapid).

**Figure 3 animals-14-02678-f003:**
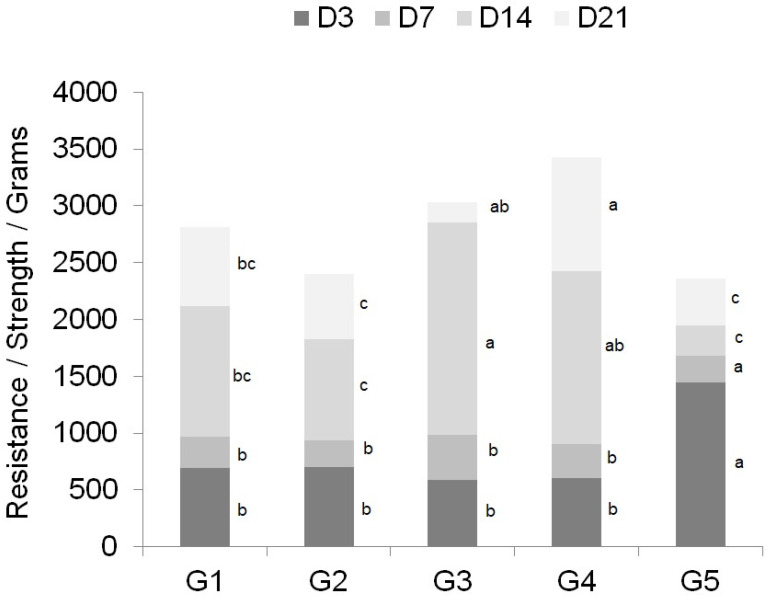
Evaluation of resistance/strength/grams of the incision line on days 3, 7, 14, and 21 after the surgical procedure. Means in different columns followed by different colors identified by lowercase letters (a, b, and c) denote significant differences between groups by the Tukey test with *p* < 0.05% probability. G1: n-2-ethyl-cyanoacrylate (Three bonder^®^); G2: n-2-ethyl-cyanoacrylate (Super bonder^®^); G3: n-2-butyl-cyanoacrylate (Histoacryl^®^); G4: n-2-octyl-cyanoacrylate (Liqui Band^®^ Rapid); and G5: nylon thread (Technofio^®^).

**Figure 4 animals-14-02678-f004:**
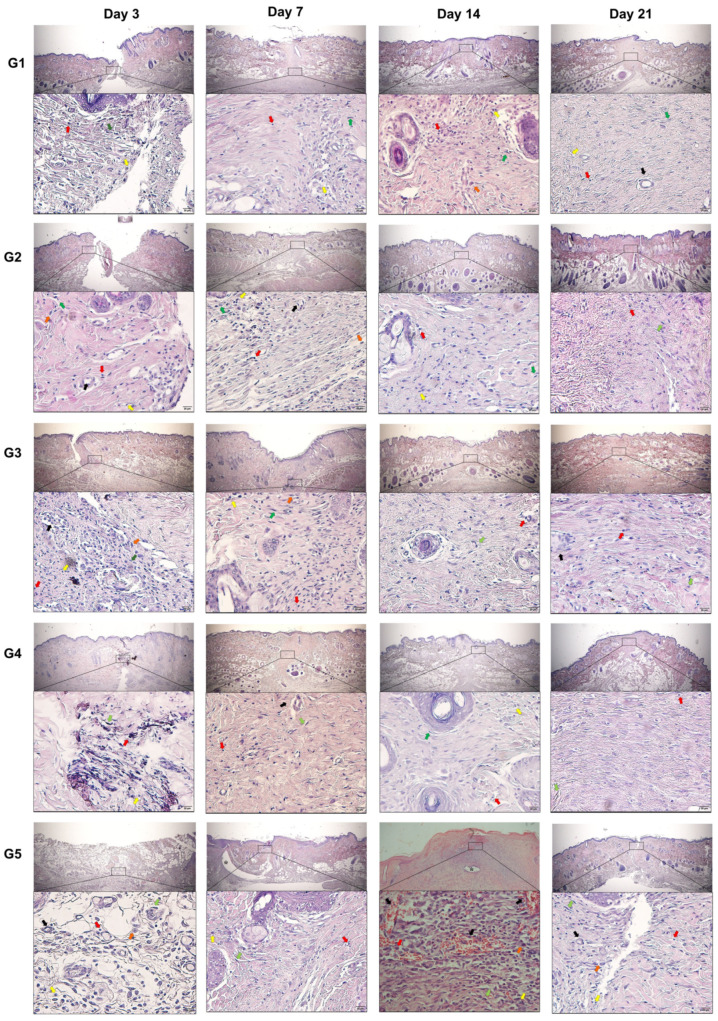
Photomicrographs of representative inflammatory infiltrate (HE stain, 40× and 400× magnification) of polymorphonuclear cells (yellow arrows), mononuclear vesicles (red arrows), blood vessels (black arrows), fibroblasts (green arrows), giant cells (orange arrows), and suture area (*). G1: n-2-ethyl-cyanoacrylate (Three bonder^®^); G2: n-2-ethyl-cyanoacrylate (Super bonder^®^); G3: n-2-butyl-cyanoacrylate (Histoacryl^®^); G4: n-2-octyl-cyanoacrylate (Liqui Band^®^ Rapid); and G5: nylon thread (Technofio^®^).

**Figure 5 animals-14-02678-f005:**
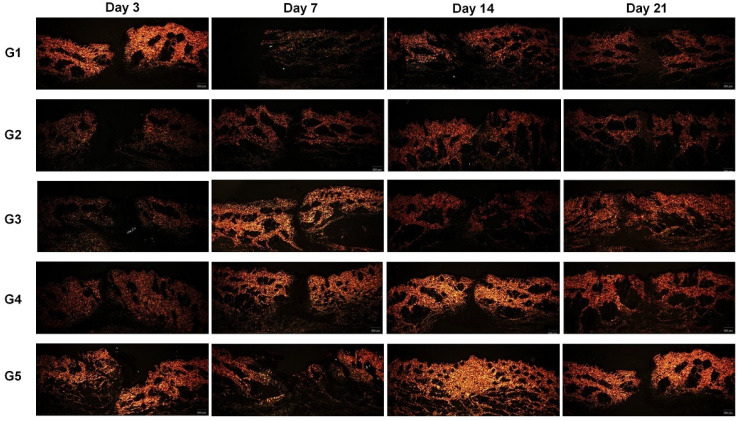
Photomicrographs stained with picrosirius red for quantification and visualization of filling of the lesion area with type III and I collagen and its evolution (40× magnification). G1: n-2-ethyl-cyanoacrylate (Three bonder^®^); G2: n-2-ethyl-cyanoacrylate (Super bonder^®^); G3: n-2-butyl-cyanoacrylate (Histoacryl^®^); G4: n-2-octyl-cyanoacrylate (Liqui Band^®^ Rapid); and G5: nylon thread (Technofio^®^).

**Figure 6 animals-14-02678-f006:**
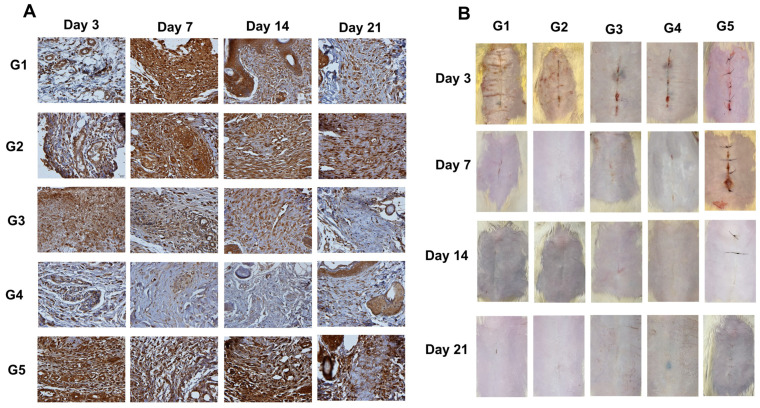
(**A**): Representative immunohistochemical photomicrographs of VEGF-α (×400 magnification). (**B**): Representative images of the wounds of the groups (*n* = 2 rats/treatment) on the 3rd, 7th, 14th, and 21st days. G1: n-2-ethyl-cyanoacrylate (Three bonder^®^); G2: n-2-ethyl-cyanoacrylate (Super bonder^®^); G3: n-2-butyl-cyanoacrylate (Histoacryl^®^); G4: n-2-octyl-cyanoacrylate (Liqui Band^®^ Rapid); and G5: nylon thread (Technofio^®^).

**Table 1 animals-14-02678-t001:** Time (days) the adhesive remained adhered.

	Day	G1	G2	G3	G4
Film formation	D0	28/28 ^a^	28/28 ^a^	28/28 ^a^	28/28 ^a^
Presence of film	D3	7/7 ^a^	7/7 ^a^	7/7 ^a^	7/7 ^a^
	D7	3/7 ^c^	3/7 ^c^	4/7 ^b^	6/7 ^a^
	D14	0/7 ^c^	0/7 ^c^	3/7 ^b^	4/7 ^a^
	D21	0/7 ^b^	0/7 ^b^	0/7 ^b^	2/7 ^a^

Means followed by distinct lowercase letters indicate significant differences between groups by the Tukey test at *p* < 0.05% probability. G1: n-2-ethyl-cyanoacrylate (Three bonder^®^); G2: n-2-ethyl-cyanoacrylate (Super bonder^®^); G3: n-2-butyl-cyanoacrylate (Histoacryl^®^); and G4: n-2-octyl-cyanoacrylate (Liqui Band^®^ Rapid).

**Table 2 animals-14-02678-t002:** Histopathological analysis of inflammatory infiltrate and granulation tissue between groups by assigning scores.

Day 3	G1	G2	G3	G4	G5
Polynuclear Cells	0.78 ± 0.20 ^b^	0.85 ± 0.69 ^b^	0.92 ± 0.34 ^b^	0.64 ± 0.24 ^b^	2.42 ± 0.93 ^a^
Mononuclear Cells	1.35 ± 0.37 ^ab^	1.50 ± 0.64 ^ab^	1.50 ± 0.40 ^ab^	1.14 ± 0.27 ^b^	2.07 ± 0.83 ^a^
Giant Cells	0.07 ± 0.19 ^a^	0.11 ± 0.20 ^a^	0.00 ± 0.00 ^a^	0.00 ± 0.00 ^a^	0.14 ± 0.24 ^a^
Granulation Tissue	0.28 ± 0.75 ^a^	0.28 ± 0.37 ^a^	0.35 ± 0.62 ^a^	0.14 ± 0.24 ^a^	0.85 ± 1.02 ^a^
**Day 7**					
Polynuclear Cells	0.85 ± 0.21 ^b^	0.94 ± 0.45 ^b^	0.64 ± 0.24 ^b^	0.57 ± 0.18 ^b^	1.98 ± 0.51 ^a^
Mononuclear Cells	1.14 ± 0.24 ^b^	1.50 ± 0.50 ^ab^	1.28 ± 0.39 ^b^	1.14 ± 0.37 ^b^	2.16 ± 0.46 ^a^
Giant Cells	0.29 ± 0.19 ^a^	0.14 ± 0.24 ^a^	0.07 ± 0.19 ^a^	0.15 ± 0.31 ^a^	0.50 ± 0.43 ^a^
Granulation Tissue	0.71 ± 0.62 ^b^	0.98 ± 0.37 ^b^	0.78 ± 1.07 ^b^	0.35 ± 0.62 ^b^	2.85 ± 1.31 ^a^
**Day 14**					
Polynuclear Cells	0.71 ± 0.43 ^b^	0.85 ± 0.39 ^b^	0.19 ± 0.48 ^b^	0.21 ± 0.26 ^b^	1.35 ± 0.63 ^a^
Mononuclear Cells	1.14 ± 0.24 ^b^	1.42 ± 0.37 ^ab^	1.14 ± 0.37 ^b^	1.25 ± 0.12 ^b^	1.95 ± 0.47 ^a^
Giant Cells	0.43 ± 0.45 ^a^	0.36 ± 0.38 ^a^	0.23 ± 0.24 ^a^	0.07 ± 0.09 ^a^	0.57 ± 0.61 ^a^
Granulation Tissue	0.71 ± 0.90 ^b^	0.71 ± 0.83 ^b^	0.14 ± 0.37 ^b^	0.14 ± 0.18 ^b^	1.48 ± 0.43 ^a^
**Day 21**					
Polynuclear Cells	0.35 ± 0.39 ^b^	0.50 ± 0.12 ^b^	0.14 ± 0.24 ^b^	0.14 ± 0.18 ^b^	0.95 ± 0.22 ^a^
Mononuclear Cells	0.85 ± 0.24 ^b^	0.98 ± 0.26 ^b^	0.71 ± 0.48 ^b^	0.35 ± 0.43 ^b^	1.64 ± 0.27 ^a^
Giant Cells	0.07 ± 0.19 ^a^	0.07 ± 0.14 ^a^	0.0 ± 0.00 ^a^	0.00 ± 0.00 ^a^	0.29 ± 0.39 ^a^
Granulation Tissue	0.23 ± 0.15 ^b^	0.35 ± 0.62 ^b^	0.07 ± 0.19 ^b^	0.05 ± 0.40 ^b^	1.14 ± 0.52 ^a^

Means followed by different lowercase letters in the same column indicate statistical differences between groups by the Mann–Whitney non-parametric test (*p* < 0.05). G1: n-2-ethyl-cyanoacrylate (Three bonder^®^); G2: n-2-ethyl-cyanoacrylate (Super bonder^®^); G3: n-2-butyl-cyanoacrylate (Histoacryl^®^); G4: n-2-octyl-cyanoacrylate (Liqui Band^®^ Rapid); and G5: 5 points of nylon thread (Technofio^®^).

**Table 3 animals-14-02678-t003:** Comparison of the percentage of the collagen maturation index by the occupation of collagen fibers of types III and I and vascular endothelial growth factor, obtained in different postoperative periods between the groups analyzed.

Day 3	G1	G2	G3	G4	G5
Type III Collagen	3.29 ± 2.49 ^b^	5.4 ± 3.6 ^ab^	3.43 ± 4.44 ^b^	3.57 ± 1.72 ^b^	10.48 ± 6.76 ^a^
Type I Collagen	2.29 ± 7.21 ^a^	2.53 ± 3.17 ^a^	2.14 ± 2.12 ^a^	3.14 ± 6.61 ^a^	2.47 ± 6.18 ^a^
VEGF-α	30.20 ± 3.84 ^bc^	24.46 ± 3.55 ^cd^	35.56 ± 7.24 ^b^	22.74 ± 2.85 ^d^	51.98 ± 10.22 ^a^
**Day 7**					
Type III Collagen	16.77 ± 8.4 ^b^	18.30 ± 4.59 ^b^	10.93 ± 1.65 ^b^	14.31 ± 5.82 ^b^	25.64 ± 4.35 ^a^
Type I Collagen	10.16 ± 2.85 ^a^	9.96 ± 4.59 ^a^	11.30 ± 1.28 ^a^	12.72 ± 1.71 ^a^	9.90 ± 2.99 ^a^
VEGF-α	46.03 ± 3.88 ^b^	58.30 ± 9.01 ^a^	30.97 ± 7.35 ^c^	27.19 ± 3.67 ^c^	53.52 ± 5.99 ^a^
**Day 14**					
Type III Collagen	14.55 ± 3.73 ^b^	16.92 ± 7.40 ^b^	12.45 ± 4.61 ^b^	12.11 ± 3.78 ^b^	34.88 ± 8.06 ^a^
Type I Collagen	26.57 ± 4.95 ^b^	28.23 ± 3.90 ^b^	30.37 ± 3.60 ^ab^	35.43 ± 2.45 ^a^	18.88 ± 4.55 ^c^
VEGF-α	41.60 ± 7.66 ^b^	45.35 ± 9.88 ^ab^	22.88 ± 3.40 ^c^	19.63 ± 6.26 ^c^	61.29 ± 12.05 ^a^
**Day 21**					
Type III Collagen	8.80 ± 4.2 ^b^	10.04 ± 6.31 ^b^	6.73 ± 1.53 ^b^	6.79 ± 4.45 ^b^	26.36 ± 6.88 ^a^
Type I Collagen	32.90 ± 4.64 ^b^	28.68 ± 5.44 ^b^	35.64 ± 3.54 ^ab^	41.65 ± 8.48 ^a^	20.78 ± 2.96 ^c^
VEGF-α	25.02 ± 4.12 ^ab^	30.74 ± 6.43 ^a^	21.89 ± 4.69 ^b^	18.90 ± 3.59 ^b^	30.19 ± 9.75 ^a^

Means followed by different lowercase letters in the same column indicate statistical differences between groups by the Mann–Whitney non-parametric test (*p* < 0.05). G1: n-2-ethyl-cyanoacrylate (Three bonder^®^); G2: n-2-ethyl-cyanoacrylate (Super bonder^®^); G3: n-2-butyl-cyanoacrylate (Histoacryl^®^); G4: n-2-octyl-cyanoacrylate (Liqui Band^®^ Rapid); and G5: 5 points of nylon thread (Technofio^®^).

## Data Availability

The original contributions presented in the study are included in the article, further inquiries can be directed to the corresponding authors.

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
