# Peer review of "Cyanoacrylate Adhesives for Cutaneous Wound Closure"

_animals, 2024, doi:10.3390/ani14182678_

Round 1

Reviewer 1 Report

Comments and Suggestions for Authors

The manuscript, "Cyanoacrylate Adhesives for Cutaneous Wound Closure," presents a well-structured and informative study comparing the effectiveness of different cyanoacrylate-based adhesives for wound closure in rats. The research is relevant and contributes valuable insights to the field of veterinary surgery. The methodology is sound, and statistical analysis presents and supports the results. However, there are a few areas where improvements could be made to enhance the overall quality of the manuscript.

  1. Mention in Figures and Tables legends about what a, b, and c represent exactly.
  2. Was the animal single-caged after the wounding/suturing
  3. What is the rationale for using Wistar rats over SD rats?
  4. Discuss the limitations of the study in more detail.
  5. Address the potential clinical implications of their findings and suggest future research directions.
  6. The conclusion could be strengthened by summarizing the study's key findings more explicitly. Also, avoid overstating the decisions based on the study's limitations.

Author Response

Caro revisor,

Gostaríamos de agradecer por revisar nosso manuscrito. Tentamos abordar todas as preocupações dos revisores apropriadamente e acreditamos que nosso artigo melhorou consideravelmente.

Pergunta 1 : Mencione nas legendas das figuras e tabelas o que a, b e c representam exatamente.

Letras minúsculas diferentes indicam diferenças estatísticas entre os grupos. Houve um erro de digitação que foi corrigido nas legendas das figuras e tabelas.

Pergunta 2 : O animal ficou em gaiola individual após o ferimento/sutura?

Os animais foram mantidos em gaiolas individuais durante todo o período experimental.

Pergunta 3 : Qual é a justificativa para usar ratos Wistar em vez de ratos SD?

Os ratos Wistar apresentam maior taxa de crescimento e ganho de peso quando comparados aos ratos SD, proporcionando maior agilidade para a execução do estudo. Além disso, são mais viáveis, com maior número de estudos conduzidos com essa linhagem de ratos.

Pergunta 4 : Discuta as limitações do estudo com mais detalhes.

As limitações do estudo foram detalhadas conforme solicitado pelo revisor.

Um parágrafo completo foi adicionado ao final da discussão com considerações sobre as limitações do estudo. Esperamos que atenda às expectativas do revisor.

Pergunta 5 : Aborde as potenciais implicações clínicas de suas descobertas e sugira futuras direções de pesquisa.

O pedido foi atendido. Um parágrafo completo foi adicionado na seção de discussão, assim como o item 6. Contribuição para a declaração de campo.

Pergunta 6 : A conclusão pode ser reforçada resumindo as principais descobertas do estudo de forma mais explícita. Além disso, evite exagerar as decisões com base nas limitações do estudo.

A conclusão foi reescrita conforme solicitado pelo revisor. Esperamos que atenda às expectativas do revisor.

Reviewer 2 Report

Comments and Suggestions for Authors

The original Article "Cyanoacrylate Adhesives for Cutaneous Wound Closure" by Inácio Silva Viana, Paula Alessandra Di Filippo, Francielli Pereira Gobbi, Rachel Bittencourt Ribeiro, Gabriel João Unger Carra, Luiza Maria Feitosa Ribeiro, Lara de Souza Ribeiro, Michelle do Carmo Pereira Rocha  and Paulo Aléscio Canola, compares the effect of four different topical adhesives based on cyanoacrylate (n-69 2-ethyl-cyanoacrylate, n-2-butyl-cyanoacrylate, n-2-octyl-cyanoacrylate) with nylon thread in the closure of an experimental cutaneous surgical wound in rats (Rattus norvegicus).

The paper is interesting and well organized, however it would be improved.

The authors state that "cyanoacrylate-based adhesives have some negative effects, including the release of cytotoxic chemicals during biodegradation." Could they measure the ROS levels released by cyanoacrylate adhesives in comparison to nylon sutures?

Do the authors have a way to verify how the expression of hyaluronic acid receptor (CD44) changes in samples treated with cyanoacrylate adhesives compared to nylon sutures?

Could the authors create a graphical abstract of the experimental setup to clarify the results?

Could Figure 3 be supplemented with the complete blood count data?"

Comments on the Quality of English Language

Minor editing of English language required.

Author Response

Dear Reviewer,

We would like to thank you for reviewing our manuscript. We have tried to address all the reviewers' concerns appropriately and believe that our paper has improved considerably.

Question 1; The authors state that "cyanoacrylate-based adhesives have some negative effects, including the release of cytotoxic chemicals during biodegradation." Could they measure the ROS levels released by cyanoacrylate adhesives in comparison to nylon sutures?

This statement was incorrectly included in the abstract, because in our study, although we sometimes observed differences between the different cyanoacrylate monomers, no histopathological characteristics of cytotoxicity were observed. For this reason, this sentence was changed.

There is no consensus in the literature regarding the occurrence of cytotoxicity of these polymers. Since our study did not evaluate ROS (by means of TBARS) and release of antioxidants (SOD), we avoided statements on this subject so as to avoid misunderstandings on this topic. However, it is known that the greater deposition, as well as the maintenance of polymorphonuclear cells during healing, promotes low O2 tension, decreased pH, and the presence of reactive nitrogen and oxygen species. This observation was added to the text.

Question 2: Do the authors have a way to verify how the expression of hyaluronic acid receptor (CD44) changes in samples treated with cyanoacrylate adhesives compared to nylon sutures?

No reports were found in the literature linking the presence of hyaluronic acid, or any other glycoprotein, to the activity of the cyanoacrylate polymer itself. However, the presence of hyaluronic acid, as well as the formation of proteoglycans, is strongly linked to the organization of granulation tissue and the restoration of tissue elasticity during the healing process.

In view of this, the paragraph was changed to include this information.

Question 3: Could the authors create a graphical abstract of the experimental setup to clarify the results?

At your request, the graphic summary has been added to the text.

Question 4: Could Figure 3 be supplemented with the complete blood count data?"

Monitoring of wounds treated with cyanoacrylate has already been carried out in some studies, and no changes were found on CBC. Given that experimental wounds are generally small, with a short healing period and in healthy animals.

Hematological monitoring is most often applied to wounds treated by second intention, as observed in healing complications in diabetic animals, ulcerated wounds, burns and infections where glucose levels, neutropenia and hemodynamic changes are observed. Based on these previous reports, CBC was not carried out during the evaluating period. Unfortunately, we are not able to provide such information.

Round 2

Reviewer 2 Report

Comments and Suggestions for Authors

The authors have provided convincing responses to the questions raised, making the manuscript now largely suitable for publication in Animals Journal.